# Differential Early Mechanistic Frontal Lobe Responses to Choline Chloride and Soy Isoflavones in an Experimental Model of Fetal Alcohol Spectrum Disorder

**DOI:** 10.3390/ijms24087595

**Published:** 2023-04-20

**Authors:** Suzanne M. de la Monte, Ming Tong, Busra Delikkaya

**Affiliations:** Departments of Pathology and Laboratory Medicine, Medicine, Neurology and Neurosurgery, Rhode Island Hospital, Lifespan Academic Institutions, The Warren Alpert Medical School of Brown University, Providence, RI 02903, USA

**Keywords:** FASD, soy, choline

## Abstract

Fetal alcohol spectrum disorder (FASD) is the most common preventable cause of neurodevelopmental defects, and white matter is a major target of ethanol neurotoxicity. Therapeutic interventions with choline or dietary soy could potentially supplement public health preventive measures. However, since soy contains abundant choline, it would be important to know if its benefits are mediated by choline or isoflavones. We compared early mechanistic responses to choline and the Daidzein+Genistein (D+G) soy isoflavones in an FASD model using frontal lobe tissue to assess oligodendrocyte function and Akt-mTOR signaling. Long Evans rat pups were binge administered 2 g/Kg of ethanol or saline (control) on postnatal days P3 and P5. P7 frontal lobe slice cultures were treated with vehicle (Veh), Choline chloride (Chol; 75 µM), or D+G (1 µM each) for 72 h without further ethanol exposures. The expression levels of myelin oligodendrocyte proteins and stress-related molecules were measured by duplex enzyme-linked immunosorbent assays (ELISAs), and mTOR signaling proteins and phosphoproteins were assessed using 11-plex magnetic bead-based ELISAs. Ethanol’s main short-term effects in Veh-treated cultures were to increase GFAP and relative PTEN phosphorylation and reduce Akt phosphorylation. Chol and D+G significantly modulated the expression of oligodendrocyte myelin proteins and mediators of insulin/IGF-1-Akt-mTOR signaling in both control and ethanol-exposed cultures. In general, the responses were more robust with D+G; the main exception was that RPS6 phosphorylation was significantly increased by Chol and not D+G. The findings suggest that dietary soy, with the benefits of providing complete nutrition together with Choline, could be used to help optimize neurodevelopment in humans at risk for FASD.

## 1. Introduction

Fetal alcohol spectrum disorder (FASD) and fetal alcohol syndrome (FAS), its severest form, are linked to maternal alcohol consumption during pregnancy [1,2] and together they represent the most common preventable causes of neurodevelopmental deficits [3,4]. FASDs’ high economic and social burdens [5] are due to the costs of caring for children with a broad range of cognitive-motor dysfunctions, including attention deficit hyperactivity disorder [6,7,8]. In the USA, FAS/FASD afflicts 0.2–1.5 per 1000 live births overall, but the rates can reach 7 or 9/1000 among women who binge drink during pregnancy [9,10]. However, in a large cross-sectional study of first-grade pupils in the USA, the estimated prevalence of FASD was as high as 98.5/1000 children [11]. FASD-associated central nervous system (CNS) pathologies [12,13,14,15] are characteristically distributed in the corpus callosum, prefrontal region, temporal lobe, and cerebellum [12,16]. In contrast to the abundant body of research on neuronal abnormalities in FASD and alcohol-related brain diseases in general, relatively little attention has been paid to white matter pathology, despite its consistent occurrence across the lifespan in both human and experimental models [12,17,18,19,20,21,22,23,24].

The mechanisms of ethanol-mediated neurodevelopmental defects have been evaluated through the systematic study of experimental models and human postmortem brains. In vitro and in vivo experiments have shown that ethanol exerts both neurotoxic and dysmetabolic effects on CNS cells and tissues [25,26,27,28,29]. Many related abnormalities in cell viability and function are associated with impairments in insulin and insulin-like growth factor type 1 (IGF-1) signaling through Akt and downstream pathways [25,30,31,32,33,34,35,36,37] at multiple levels within the cascade, beginning with the insulin/IGF-1 receptors [31,34,38,39,40,41]. Attendant reductions in signaling through insulin receptor substrate (IRS) proteins and Akt lead to increased activation of glycogen synthase kinase 3β (GSK-3β) [31,34,38,39,40,41], which is inhibitory to CNS cell survival, metabolism, and function [30,35,37,42,43]. Consequences include reduced expression of target proteins that are needed for various functions in both immature and mature brains [31,44,45,46,47,48,49].

Multiple studies of white matter pathology have demonstrated loss of volume due to the depletion of myelin and axons [12,17,24,50,51], and oligodendrocyte dysfunction manifested by impaired survival due to increased apoptosis [52], altered myelin-associated protein expression [48,49] and shifts in sphingolipid and phospholipid profiles [53,54,55,56]. Alcohol-related shifts in brain lipid profiles linked to neurobehavioral dysfunction may be due to dysregulated metabolism. Correspondingly, partial normalization of ethanol-induced neurobehavioral deficits was achieved by treatment with myriocin, which reduced the white matter accumulations of toxic ceramides and increased sphingomyelin and sulfatide [53]. However, ethanol’s neurotoxic, neurodevelopmental, and neurodegenerative effects in white matter have been mechanistically linked to impairments in insulin/IGF-1 signaling through Akt pathways [57]. Awareness of the contributions of impaired insulin/IGF-1 signaling in alcohol-related brain diseases led to the concept that the deleterious effects of ethanol could be minimized or prevented by therapeutic targeting with insulin sensitizer agents. In support of this concept were the findings that peroxisome proliferator-activated receptor (PPAR) agonists, which have both insulin/IGF-1 sensitizing and antioxidant effects [58,59,60] were protective against the adverse consequences of ethanol exposure in both cortical [61] and white matter structures. Mechanistically, the PPAR agonist treatments normalized neurobehavioral function and neuroglial cellular protein expression, reduced organ/tissue pathology, and restored critical aspects of insulin signaling through Akt [61,62].

Despite promising results from pre-clinical experiments, the prospect of translating the PPAR agonist approach to human clinical trials is limited due to unknown risks to the maternal-placental-fetal unit. This concern led us to consider an alternative natural insulin-sensitizer and antioxidant food source, namely dietary soy [63,64,65]. Soy isoflavones support insulin responsiveness and combat insulin-resistant disease states [63,65,66,67,68,69]. Recently, we demonstrated that dietary soy could prevent chronic ethanol consumption-associated impairments in spatial learning and memory on Morris water maze tasks in an adolescent rat model [70] and that maternal consumption of dietary soy prevented both placental and craniofacial phenotypic pathologies in an experimental model of FASD [71]. However, a potential confounder in the data interpretation is that dietary soy contains substantial amounts of choline [66,67].

Choline, an essential nutrient [72], is required for the synthesis of acetylcholine and major phospholipids, including phosphatidylcholine, lysophophatidylcholine, choline plasmalogen, sphingomyelin, cell-membrane signaling, lipid transport, and methyl group metabolism [72]. Multiple studies have shown that choline supplementation benefits neurocognitive functions at risk following developmental alcohol exposures in experimental animals [73,74,75] and humans [76,77]. Choline was also shown to be neuroprotective in models of neuronal damage and to aid in normalizing the development of structures and neurobehavioral functions damaged in FASD [78,79,80]. In children with FASD, choline treatment improved non-verbal intelligence, visual-spatial skills, working memory capacity, verbal memory, and rates of attention deficit hyperactivity disorder [76].

Nonetheless, one argument favoring the use of soy over choline for FASD is that soy is a natural whole food that provides healthful complete protein and can fully replace cow’s milk [81]. Soy protein isolate is a high-quality product of soy protein and contains three main proteins: β-conglycinin, glycinin, and lipophilic proteins, which together have an amino acid composition that closely approximates animal proteins [81]. In addition, soy contains isoflavone phytochemicals, including daidzein, genistein, and glycitein [82] which have insulin sensitizing and antioxidant effects [63,65,69], and is an abundant source of choline [66]. The isoflavone concentration in soy is 1.5 mg/g [83] and the choline concentration is 120 mg/100 g of soybeans [84]. Therefore, the use of dietary soy would provide the additional benefit of correcting nutritional deficiencies that often accompany inadequate choline as well as other micronutrient intake during pregnancy in socioeconomically deprived environments [85]. These considerations led to the present study, which was designed to determine if the therapeutic effects of dietary soy were mediated by or distinct from those of choline. The study design utilized frontal lobe slice cultures from a binge FASD rat model to compare the therapeutic effects of choline with those mediated by the chemoprotective dietary soy isoflavones, Daidzein and Genistein [83]. Furthermore, in light of prior evidence that dietary soy can positively impact normal brain development and function [70,71,86], it was of interest to also compare the responses to soy and choline in control brain samples.

## 2. Results

The experiment was designed to examine the very early effects of 3rd trimester-equivalent binge alcohol exposures to gain a better understanding of the initiating molecular and biochemical responses that precede the consistent development of FASD-associated neurodevelopmental and neurobehavioral defects together with significant impairments in insulin and IGF-1 signaling through Akt pathways in the brain, including white matter [37,61,87,88,89,90]. In essence, the long-term effects of binge alcohol exposure have been established, but little is known about the initial pathologies that ultimately progress to white matter abnormalities in FASD. The second main focus of the investigation was how Chol or D+G impact the expression of glial and stress proteins and signaling through upstream, middle level, and downstream components of the Akt-mTOR pathway. Although the long-term effects have been established, the initial mediators of eventual therapeutic remediation have not been evaluated in white matter. Moreover, it was of interest to understand how Chol and D+G impact control of cerebral white matter, given the strong evidence that Chol and dietary soy have positive effects on the development and function of both normal and diseased brains [64,70,71,73,74,75,80,86]. Finally, the in vivo–ex vivo study design combined the strengths of a well-characterized in vivo model with the advantages of controlling the brain exposure levels of Chol and D+G. The CNS bioavailability of parenterally administered D+G has not been studied. Thus, in these studies, the goal was to draw comparisons between Chol and D+G based on known levels of tissue exposure.

### 2.1. Oligodendrocyte Proteins

MAG1, MOG, MBP, PLP, PDGFR-α, and GALC are expressed in white matter oligodendrocytes at varying stages of differentiation [91]. Their functions are summarized in Appendix A. These analyses were performed because, although previous studies have characterized the long-term effects of chronic or repeated binge ethanol exposures on white matter degeneration or developmental pathology in relation to neurobehavioral impairments [37,75], altered survival [52], and the expression of oligodendrocyte genes and proteins [87,92], little is known about early responses to ethanol or potential neuroprotective treatments.

MAG1: MAG1, a relatively minor component of myelin, helps maintain myelin-axon spacing through its interactions with specific neuronal gangliosides and controls myelin formation [93,94]. Ethanol and treatment (Chol or D+G) each had significant effects on MAG1 expression; however, there were no significant ethanol x treatment interactive effects (Table 1). The main inter-group differences were that MAG1 expression was significantly reduced by ethanol in Chol-treated cultures and similarly elevated in control and ethanol-exposed D+G relative to vehicle- or Chol-treated cultures. Therefore, D+G significantly increased MAG1, whereas Chol did not (Figure 1A).

MOG: MOG, a surface marker of oligodendrocyte maturation, is expressed very late in development and parallels myelination [95]. A two-way ANOVA showed that MOG was significantly modulated by treatment (Chol or D+G) and ethanol x treatment interactions (Table 1). The main post-hoc test results were that MOG expression was significantly higher in control–Chol, control–D+G, and ethanol–D+G than in control–Veh, ethanol–Veh, and ethanol–Chol (Figure 1B). Therefore, D+G significantly increased MOG in ethanol-exposed cultures relative to vehicle-treated cultures, whereas Chol did not.

MBP: MBP, the second most abundant protein in CNS myelin, aids in the formation and stabilization of myelin membranes [96]. MBP was not significantly modulated by ethanol, treatment (Chol/D+G), or ethanol x treatment interactions (Table 1). Correspondingly, the mean levels of MBP were similar for all groups, irrespective of ethanol exposure and treatment (Figure 1C).

PLP: PLP, the most abundant CNS myelin protein [97], was significantly modulated by treatment (Chol or D+G) (Table 1). The main finding was that ethanol–Chol, ethanol–D+G, and control–D+G significantly reduced PLP relative to ethanol–Veh (Figure 1D).

PDGFR-α: PDGFR-α, a cell-surface tyrosine kinase receptor, is expressed in immature and mature brains, including in neural progenitor cells, neurons, astrocytes, oligodendrocytes, and vascular elements. In oligodendrocyte progenitor cells, PDGFR-α mediates survival, growth, and maintenance [98,99]. PDGFR-α expression was significantly modulated by treatment (Chol or D+G) (Table 1). PDGFR-α was significantly elevated in Chol- and D+G-treated mice relative to Veh, irrespective of ethanol exposure (Figure 1E).

GALC: This dominant type of glycosphingolipid in CNS myelin functions in intracellular communication and cellular development and gets metabolized to sulfatide [100]. GALC expression was significantly modulated by treatment (Chol or D+G) (Table 1). GALC expression progressively and significantly increased from the lowest levels in control–Veh and ethanol–Veh to higher levels in control–Chol and ethanol–Chol, with further significant increases due to D+G, irrespective of ethanol exposure (Figure 1F).

### 2.2. Astrocytes and Stress Molecules

Included in this cluster are GFAP, Ubiquitin, 8-OHdG, and HNE.

GFAP: GFAP is a major astrocyte intermediate filament cytoskeletal protein that responds to injury [101]. Statistical trend effects (0.05 ≤ *p* ≤ 0.10) for treatment and ethanol x treatment interactions were detected by two-way ANOVA (Table 1). The most notable observation was that the ethanol–Veh-associated elevation in GFAP was abolished by Chol or D+G (Figure 2A).

Ubiquitin: Ubiquitin has roles in non-lysosomal, energy-dependent protein degradation and is particularly important for targeted removal of misfolded and aggregated unwanted proteins, but it also mediates synaptic functions [102]. A two-way ANOVA demonstrated that ubiquitin was significantly modulated by treatment (Chol or D+G) (Table 1). D+G significantly increased ubiquitin in control cultures relative to vehicle- and Chol-treated control and ethanol-exposed cultures (Figure 2B). D+G also increased ubiquitin in ethanol relative to the control–vehicle. In contrast, Chol did not significantly modulate ubiquitin expression relative to the vehicle.

8-OHdG: 8-OHdG is a major product of DNA oxidation mediated by reactive oxygen species attack on guanine bases, marking impaired growth during development [103,104]. A two-way ANOVA showed that 8-OHdG immunoreactivity was significantly modulated by treatment (Chol or D+G), but not by ethanol or ethanol x treatment interactions (Table 1). The post-hoc tests demonstrated similarly elevated 8-OHdG in control–D+G and ethanol–D+G cultures, with significant differences from corresponding Veh- and Chol-treated cultures (Figure 2C), and significant differences between ethanol–D+G and controls in the Veh and Chol groups.

HNE: HNE is a major, α,β-unsaturated aldehyde product of n-6 fatty acid oxidation and a lipid peroxidation end-product that functions as a second messenger of oxidative/electrophilic stress [105]. HNE modulates cell survival/death via ER stress induction and promotes cell death via apoptosis [105]. HNE levels were significantly affected by ethanol and treatment (Table 1). The mean levels of HNE were significantly reduced by Chol in ethanol cultures and by D+G in both control and ethanol cultures. The main findings were that both Chol and D+G reduced HNE in the ethanol-exposed cultures relative to the Veh-treated cultures, and D+G reduced HNE in the control cultures whereas Chol did not.

### 2.3. Upstream Insulin/IGF-1 Signaling

Insulin and IGF-1 receptors (Insulin-R and IGF-1R) are the most upstream components of the pathway. Tyrosine phosphorylation and attendant activation of their intrinsic-receptor tyrosine kinases lead to tyrosine phosphorylation of IRS-1, which interacts with adaptor molecules to mediate a broad array of cellular functions [106,107,108]. Downstream signaling through PI3K-Akt-mTOR [109,110,111] leads to activation of mTORC1 [111]. However, serine phosphorylation via a negative feedback loop from mTORC1 negatively regulates IRS1 [111]. The two-way ANOVA test results for Insulin-R, IGF-1R, and IRS-1 are provided in Table 2, and the graphed data with significant post-hoc differences are depicted in Figure 3.

Insulin-R: Ethanol had a statistical trend effect on ^pYpY1162/1163^-Insulin-R. Treatment with Chol or D+G had significant effects on Insulin-R and the relative levels of Insulin-R phosphorylation (p/T) (Table 2). D+G significantly reduced Insulin-R in control and ethanol-exposed cultures relative to ethanol–Veh (Figure 3A), but increased p/T-Insulin-R, i.e., signaling, relative to corresponding control and ethanol-exposed Veh-treated cultures (Figure 3C). p/T-Insulin-R was also higher in ethanol–D+G than control–Veh. In contrast, Chol had no significant effects on Insulin-R, ^pYpY1162/1163^-Insulin-R, or p/T-Insulin-R (Figure 3A–C).

IGF-1R: Ethanol significantly impacted the ^pYpY1135/1136^-IGF-1R levels and had a trend effect on p/T-IGF-1R. Treatment (Chol or D+G) significantly modulated the IGF-1R and p/T-IGF-1R (Table 2). D+G significantly reduced IGF-1R relative to Veh, irrespective of ethanol exposure (Figure 3D). In addition, IGF-1R expression in control–Chol was significantly reduced relative to control–Veh. ^pYpY1135/1136^-IGF-1R in ethanol–Chol, control–D+G, and ethanol–D+G were significantly elevated relative to control–Veh (Figure 3E). D+G significantly increased p/T-IGF-1R in control and ethanol-exposed cultures relative to control–Veh and ethanol–Veh, and control-Chol. Intermediate effects of Chol were manifested by significantly higher levels of p/T-IGF-1R relative to control–Veh and lower or similar levels compared with D+G (Figure 3F).

IRS-1: The IRS-1 protein, ^pS636^-IRS-1, and p/T-IRS-1 were not significantly modulated by ethanol, treatment with Chol or D+G, or ethanol x treatment interactions. Correspondingly, the mean levels of IRS-1, ^pS636^-IRS-1, and p/T-IRS-1 were relatively uniform across all groups (Figure 3G–I).

### 2.4. Akt Pathway (Table 3 and Figure 4)

Akt, PTEN, GSK-3α, and GSK-3β have roles in regulating downstream mTOR signaling. Tyrosine phosphorylated IRS-1 engages the p85 subunit of PI3 Kinase, whose kinase activates PDK, leading to Serine phosphorylation of Akt (^pS473^-Akt). The activated Akt kinase then phosphorylates TSC1/2, releasing its inhibitory check on mTOR signaling [112,113]. Akt kinase, activated by Insulin-R and IGF-1R tyrosine kinases, stimulates growth, survival, and metabolism [114,115]. PTEN is a potent inhibitor of PI3K-Akt and mTOR [116,117]. GSK-3β- and CK2-mediated phosphorylation of PTEN slows its proteasome degradation and thereby negatively impacts mTOR [118,119,120]. Evidence suggests that GSK-3α negatively regulates Akt via T312 phosphorylation, and that IKKi-mediated IL-1-induced GSK-3α phosphorylation at S21 inactivates GSK-3α, and thereby increases Akt activation of mTOR [121]. GSK-3β’s multi-targeted roles as upstream and downstream negative regulators of mTOR have been well-established, along with insulin/IGF-1/IRS-Akt inhibition of the Ser/Thr protein kinase [118].

Akt: Ethanol significantly affected ^pS473^-Akt and had a trend effect on p/T-Akt. Treatment (Chol or D+G) had a trend effect on ^pS473^-Akt and a significant effect on p/T-Akt. The mean levels of Akt did not significantly differ among the groups (Figure 4A). Ethanol significantly reduced ^pS473^-Akt in Veh-treated cultures. In contrast, D+G normalized ^pS473^-Akt in ethanol-exposed cultures, rendering the levels similar to all control groups and significantly higher than in ethanol–Veh and ethanol–Chol (Figure 4B). D+G also normalized p/T-Akt in ethanol-exposed cultures and significantly increased the mean level relative to ethanol–Veh and ethanol–Chol (Figure 4C). In contrast, Chol did not rescue ethanol’s inhibitory effects on ^pS473^-Akt or p/T-Akt.

**Figure 4 ijms-24-07595-f004:**
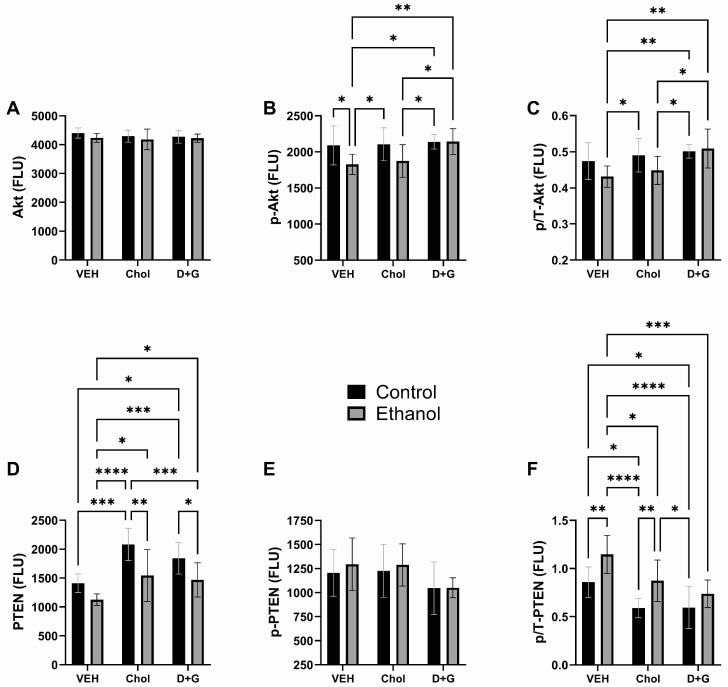
Ethanol and treatment **(**Chol and D+G) effects on intermediate signaling molecules in the Insulin/IGF-1-Akt-mTOR pathway. (**A**) Akt, (**B**) ^pS473^-Akt, (**C**) p/T-Akt, (**D**) PTEN, (**E**) ^pS380^-PTEN, (**F**) p/T-PTEN. Graphs depict the mean ± SD of (N = 6 cultures/group; FLU = fluorescent light units). Inter-group differences were analyzed by a two-way ANOVA (see Table 3) with post-hoc multiple comparisons using Tukey tests. * *p* < 0.05; ** *p* < 0.01; *** *p* < 0.001; **** *p* < 0.0001.

**Table 3 ijms-24-07595-t003:** Effects of In Vivo Ethanol Exposure and Ex Vivo Interventive Treatments on Akt, PTEN, and GSK-3 Signaling Mediators in Frontal Lobe Slice Cultures.

Protein	Ethanol Factor		Treatment Factor		Ethanol x Treatment Interaction	
	F-Ratio	*p*-Value	F-Ratio	*p*-Value	F-Ratio	*p*-Value
Akt	2.214	N.S.	0.491	N.S.	0.262	N.S.
p-Akt	**6.113**	**0.01**	** *2.915* **	** *0.0697* **	1.599	N.S.
p/T-Akt	** *3.434* **	** *0.074* **	**4.977**	**0.014**	1.432	N.S.
PTEN	**17.94**	**0.0002**	**11.831**	**0.0002**	0.616	N.S.
p-PTEN	0.413	N.S.	** *2.928* **	** *0.0689* **	0.097	N.S.
p/T-PTEN	**16.39**	**0.0003**	**12.17**	**0.0001**	0.527	N.S.
GSK-3α	1.153	N.S.	**9.199**	**0.0008**	0.482	N.S.
p-GSK-3α	0.039	N.S.	0.119	N.S.	0.128	N.S.
p/T-GSK-3α	0.166	N.S.	0.229	N.S.	0.232	N.S.
GSK-3β	0.413	N.S.	**8.562**	**0.001**	0.378	N.S.
p-GSK-3β	0.684	N.S.	** *3.108* **	** *0.059* **	0.598	N.S.
p/T-GSK-3β	** *3.591* **	** *0.068* **	**12.37**	**0.0001**	0.014	N.S.

Immunoreactivity was measured with 11-Plex Akt/mTOR total protein and phosphoprotein magnetic bead-based assays. The table lists the two-way ANOVA test results (F-Ratios and *p*-Values) for ethanol, treatment (Chol or D+G), and ethanol x treatment interactive effects on Akt, PTEN, GSK-3α, GSK-3β, ^pS473^-Akt, ^pS380^-PTEN, ^pS21^-GSK-3α, and ^pS9^-GSK-3β, and the calculated relative phosphorylation (p/T) levels of each protein. DF for all tests: interaction DFn, DFd (2, 30); treatment DFn, DFd (2, 30); exposure DFn, DFd (1, 30). The bold (non-italicized) font highlights significant results. Bold italics highlight statistical trends. Abbreviations: R = receptor; p = phosphorylated; N.S. = not statistically significant. N = 6 rats/group with 3 replicate cultures per rat.

PTEN: The PTEN protein and p/T-PTEN were significantly modulated by ethanol and treatment with Chol or D+G. A statistical trend effect of treatment was observed for ^pS380^-PTEN (Table 3). PTEN levels were lowest in Veh-treated cultures and significantly elevated in control–Chol and control–D+G relative to control–Veh and ethanol–Veh (Figure 4D). Ethanol significantly muted the Chol- and D+G-associated increases in PTEN, although the mean levels were still significantly higher than ethanol–Veh. The mean levels of ^pS380^-PTEN varied but did not differ significantly among the groups (Figure 4E). The mean levels of p/T were higher following ethanol exposure, irrespective of treatment (Figure 4F). However, Chol- and D+G significantly reduced p/T-PTEN in both control and ethanol-exposed cells relative to corresponding Veh-treated cells (Figure 4F).

GSK-3α: Ethanol and ethanol x treatment interactions had no significant or trend effects on GSK-3α, ^pS21^-GSK-3α, or p/T-GSK-3α. Treatment (Chol or D+G) significantly impacted GSK-3α protein but not ^pS21^-GSK-3α or p/T-GSK-3α. GSK-3α expression was highest in control–Veh followed by ethanol–Veh. Choline treatment of control and ethanol-exposed cultures significantly reduced GSK-3α relative to control–Veh (Figure 5A). D+G significantly reduced GSK-3α in control and ethanol-exposed mice relative to Veh, irrespective of ethanol exposure (Figure 5A). There were no significant effects of ethanol or treatment on ^pS21^-GSK-3α or p/T-GSK-3α (Figure 5B,C). In essence, the Chol and D+G treatments would have net reduced GSK-3α activity, enhancing mTOR relative to Veh.

GSK-3β: Ethanol had a statistical trend effect on p/T-GSK-3β. Treatment (Chol or D+G) had significant effects on GSK-3β protein and p/T-GSK-3β, and a trend effect on ^pS9^-GSK-3β (Table 3). GSK-3β was lowest in control and ethanol-exposed D+G treated samples, resulting in broadly significant differences from the Veh- and Chol-treated cultures (Figure 5D). ^pS9^-GSK-3β was significantly lower in ethanol–D+G than in Chol-treated control and ethanol-exposed cultures (Figure 5E). p/T-GSK-3β progressively increased from Veh to Chol to D+G, with modestly lower levels associated with ethanol exposure. D+G significantly increased p/T-GSK-3β in control and ethanol cultures relative to Veh. Intermediate responses to Chol were manifested by significantly higher p/T-GSK-3β in control–Chol relative to ethanol–Veh but lower levels in control–Chol and ethanol–Chol relative to control–D+G or ethanol–D+G (Figure 5F).

### 2.5. mTOR Pathway

mTOR signaling is mediated by two main complexes termed mTORC1 and mTORC2, in which mTOR functions as the catalytic subunit at the center. The mTORC1 complex is sensitive to rapamycin, composed of mTOR, Raptor, mammalian lethal with SEC13 protein 8 (mLST8), proline-rich Akt substrate of 40 kEa (PRAS40), and DEP domain-containing mTOR-interacting protein (Deptor), and localized in endosomal and lysosomal membranes. mTORC2 is insensitive to rapamycin, composed of mTOR bound to Rictor, Protor-1/2, mammalian stress-activated MAP kinase interacting protein 1 (mSIN1), mLST8, and Deptor, and is associated with the plasma and ribosomal membranes [122]. mTORC1 is regulated broadly by stimuli such as growth factors (Insulin/IGF-1), stressors including hypoxia and nutrients, and energy status, while mTORC2 is largely regulated by growth factor stimulation. Non-phosphorylated TSC1/2 constitutively inhibits mTOR signaling. Akt kinase phosphorylation inactivates TSC2, releasing its brake on mTORC1/2 signal transduction [123,124]. mTORC1/2′s broad cellular effects on promoting mitochondrial function, cytoskeletal organization, cell migration, dendrite formation, glial differentiation, and lipid and protein metabolism while inhibiting autophagy are mediated by serine phosphorylation with attendant kinase activation [125,126,127]. P70S6K, a major substrate for Akt-activated mTORC1 via serine phosphorylation [127], regulates mRNA translation, modulates cell cycle progression, cell survival, and cell size, and inhibits apoptosis by inhibiting mitochondrial BAD via phosphorylation and activation [128,129]. ^pS235/236^-RPS6 is the active form of S6 and the functional readout of mTORC1 activation of p70S6K [130].

TSC2: Ethanol had a significant effect on p/T-TSC2, whereas treatment with Choline or D+G significantly impacted TSC2 protein, ^pS939^-TSC2, and p/T-TSC2 (Table 4). D+G treated control and ethanol-exposed cultures had the lowest mean levels of TSC2, with broadly significant differences from Veh- and Chol-treated cultures (Figure 6A). ^pS939^-TSC2 was significantly lower in ethanol–D+G compared with control and ethanol-exposed Veh- or Chol-treated cultures (Figure 6B). p/T-TSC2 increased progressively from Veh to Chol to D+G, but with less robust responses in the ethanol–Chol group (Figure 6C). Consequently, in control–D+G, p/T-TSC2 was significantly higher than in the Veh and Chol-treated cultures, irrespective of ethanol exposure, and p/T-TSC2 in ethanol–D+G was significantly higher than ethanol–Chol, control–Veh, and ethanol–Veh. The responses to Chol were intermediate between Veh and D+G (Figure 6C). Altogether, D+G was more effective than Chol in mediating relative phosphorylation and inhibition of TSC2.

mTOR: Ethanol had significant effects on ^pS2448^-mTOR and p/T-mTOR, while treatment (Chol or D+G) significantly impacted mTOR, ^pS2448^-mTOR, and p/T-mTOR expression. In addition, an ethanol x treatment interactive statistical trend effect was observed for p/T-mTOR (Table 4). The mean levels of mTOR progressively declined from the Veh to Chol to D+G treatments, resulting in significantly lower mTOR expression in D+G- relative to Veh, irrespective of ethanol exposure (Figure 6D). Chol’s intermediate responses resulted in significantly lower mTOR expression in control and ethanol-exposed groups relative to corresponding Veh treatments and higher levels of mTOR in control–Chol than control–D+G. Regarding ^pS2448^-mTOR and p/T-mTOR, the main finding was that the ethanol–Chol levels were significantly higher than in all other groups (Figure 6E,F).

p70S6K: Ethanol had a statistical trend effect on p/T-p70S6K, while treatment (Chol or D+G) had significant effects on the levels of p70S6K and p/T-p70S6K (Table 4). The mean levels of p70S6K declined progressively from Veh to Chol to D+G treatments, resulting in significantly reduced expression in D+G relative to Veh, irrespective of ethanol exposure (Figure 7A). In addition, p70S6K expression was significantly lower in ethanol–D+G than in control–Chol. The mean levels of ^pT412^-p70S6K were similar across all groups (Figure 7B). The treatment group-wise shifts in p/T-p70S6K were opposite those of the p70S6K protein. The Chol- and D+G-associated increases in p/T-p70S6K differed significantly from control–Veh. Ethanol–D+G also was significantly increased relative to ethanol–Veh (Figure 7C).

RPS6: Treatment (Chol or D+G) significantly impacted RPS6, ^pS235/236^-RPS6, and p/T-RPS6, while ethanol and ethanol x treatment interactions had no significant or trend effects on these indices (Table 4). D+G reduced RPS6 expression, resulting in significantly lower levels in ethanol–D+G relative to Veh- or Chol-treated control and ethanol-exposed cultures, and lower levels in control–D+G relative to Veh, irrespective of ethanol exposure (Figure 7D). Chol significantly increased ^pS235/236^-RPS6 (Figure 7E) and p/T-RPS6 (Figure 7F) in the control group relative to all other groups except ethanol–Chol and in the ethanol-exposed group relative to control–D+G or ethanol–D+G. This selective Chol-associated modulation of RPS6 phosphorylation and relative phosphorylation was distinct from its otherwise intermediate effects on myelin proteins, HNE, signaling proteins, phosphorylated signaling proteins, and relative levels of signaling proteins.

## 3. Discussion

FASD comprises a group of developmental abnormalities that have been causally linked to excessive prenatal alcohol exposure, and although preventable via abstinence, social, cultural, and educational forces continue to challenge the effective implementation of public health measures. However, promising research in experimental models and humans has demonstrated the potential value of choline supplementation for reducing FASD [73,74,75,76,77]. The basis for this approach is rooted in the requirements for adequate choline intake and reserves to support normal neurodevelopment, energy metabolism, and brain functions, including the generation of acetylcholine [131,132], yet its deficiency is common [72,85,133,134,135,136].

Independent studies focused on underlying mechanisms of alcohol-mediated neurodevelopmental defects identified inhibition of insulin and IGF-1 signaling as critical mediators of FASD-related pathologies, including neuronal loss, impaired neuronal migration, reductions in energy metabolism and mitochondrial function, increased oxidative stress, and deficits in cognitive-motor functions [25,30,31,32,33,34,35,36,37]. Furthermore, several preclinical studies showed that many adverse effects of prenatal alcohol exposure can be prevented or reduced by treatment with peroxisome proliferator-activated receptor (PPAR) agonists that have both insulin sensitizer and antioxidant actions [61,137,138,139]. Importantly, PPAR agonists that target both the delta and gamma receptors were found to be effective in preventing or reducing permanent neurobehavioral and motor dysfunctions in alcohol exposure and other models with brain insulin resistance [61,138,139,140,141]. However, the use of PPAR agonists in pregnant humans, neonates, and young children is potentially problematic due to unknown long-term adverse effects. Therefore, our research progressed to testing the hypothesis that dietary soy, a natural food that has known insulin-sensitizer and antioxidant actions [63,68,142] could be used to prevent or harm-reduce the effects of FASD.

Two recent preclinical studies support the use of dietary soy as a strategy for preventing the long-term adverse effects of excessive alcohol exposure during development. In an adolescent model, dietary soy replacement of casein in the standard rodent diet prevented long-term neurocognitive and motor dysfunctions linked to chronic heavy ethanol consumption [70]. In the second study, utilizing a chronic gestational alcohol exposure model, dietary soy prevented FASD-associated impairments in placentation, craniofacial dysmorphic features, and intrauterine growth restriction [71]. Mechanistically, dietary soy enhanced insulin and IGF-1 signaling through metabolic, growth, and antioxidant pathways required for placentation and fetal growth [70,71].

One potential confounder concerning the dietary soy studies is that soy contains abundant choline, and therefore, the observed therapeutic effects could have been mediated by choline rather than soy. On the other hand, there is ample evidence that soy isoflavones have positive effects on insulin-resistance disease states [63,66,67,68], which hypothetically would include FASD. Since the benefits of choline supplementation during pregnancy or postnatal development have not been universally consistent or robust and have been variably confounded by compromised maternal nutritional, socioeconomic, and educational statuses [143,144], further research is needed to assess the relative value of alternative approaches, including the provision of whole foods rather than micronutrients. This study was designed to mechanistically compare the supportive effects of choline with those of major soy isolate bioactive constituents, namely daidzein and genistein isoflavones, in an established model of FASD. Daidzein and genistein were administered together (D+G) because exploratory studies showed greater efficacy with their combined versus individual use for supporting CNS tissue viability and function in vitro.

The experimental model of FASD was generated by administering i.p. injections of ethanol on P3 and P5, which is equivalent to 3rd trimester human exposures. Previous studies demonstrated this approach to be effective for producing robust long-term impairments in cognitive-motor functions, brain development, neuronal and glial protein/gene expression, and insulin/IGF-1 signaling [37,87,145], similar to the abnormalities observed with chronic prenatal exposures via the pregnant dam’s diet [42,45,146]. Although intragastric administration enables binge alcohol exposures [147,148,149,150], the alternative intraperitoneal injection approach is recognized by the alcohol research field [151] and has been successfully utilized by investigators over at least the past two decades [152,153,154,155]. Moreover, even maternal i.p. alcohol administration has been used to produce binge alcohol-mediated FASD [89,90].

Frontal lobe slice cultures were studied due to the relative abundance of white matter in coronal sections and the importance of white matter targeting in FASD. The use of slice cultures was also beneficial because physiological responses within the intact three-dimensional tissue architecture, in which all cell types are represented, resemble the in vivo state more closely than those that occur with isolated, cultured cells or cell lines. The early postnatal binge ethanol exposure model corresponds to human 3rd-trimester equivalent binge drinking [156,157], and was especially suited for this research because previous studies showed that long-term FASD-associated cognitive-motor deficits, brain structural pathology, and molecular and biochemical pathologies persist through adolescence [147,158].

The investigations focused on early responses (after 72 h of treatment) in relation to oligodendrocyte myelin proteins, stress molecules, and metabolic signaling through the Insulin/IGF-1-IRS1-Akt/mTOR pathways. Previous analyses of alcohol’s effects on insulin signaling attended to insulin/IGF-1/IRS-Akt-GSK-3β, which broadly impact CNS cell viability/survival, metabolic functions, neuronal growth and plasticity, and cellular adhesion and migration [30,31,32,34,36,37,42,45]. The expanded analysis of downstream mTOR-related signaling attends to their importance in relation to white matter development and myelination [159,160,161] and the impact of ethanol on oligodendrocyte function in immature white matter [12,52,54,57,61,162,163]. Ex vivo treatment with D+G was conducted to precisely control the treatment of brain tissue because of the effectiveness with which parenteral, intra-gastric, or maternal delivery of D+G reaches the brain, and whether the biodistribution differs for ethanol compared with control animals is unknown. A direct application approach for treating ethanol-exposed hippocampal slice cultures was used to examine choline chloride’s short-term effects on long-term depression [80].

The rationale for conducting these studies at a very early timepoint, prior to the anticipated development of significant FASD effects, was threefold: (1) molecular and biochemical changes precede structural and neurobehavioral abnormalities; (2) the responses to potential therapeutic interventions that have already proven effective in long-term studies will likely begin early and “re-set” the trajectory; and (3) since both choline and dietary soy have positive effects on normal brain development and function, it was of interest to examine early responses in control samples to increase understanding of their mechanisms of action in non-disease states. The early analytical studies were informative because the findings suggest that ethanol initially impairs mid-level and downstream signaling through Akt/mTOR with associated increased lipid peroxidation and inhibition of MAG1 expression, and therefore the upstream impairments of insulin and IGF-1 signaling may develop later. In addition, the early timepoint studies showed that Chol or D+G quickly impacted HNE, MAG1, MOG, PDGFR-α, GALC, and both protein and phospho-protein signaling molecules at multiple levels within the Insulin/IGF-1-Akt-mTOR pathway. The limitations of short-term evaluations will be addressed by further long-term experiments to directly compare the in vivo molecular and biochemical therapeutic effects of choline versus soy isolate constituents and the potential effects of sex. The remaining discussion dissects the outcomes and significance of each component of the experiment.

Oligodendrocyte Proteins: The most striking differences between the present ex vivo observations and previously reported results of direct brain tissue analysis pertained to the minimal early effects that ethanol had on oligodendrocyte protein expression in vehicle-treated cultures versus the significant ethanol-induced alterations detected at much later in vivo time points, i.e., 4 weeks or longer [92,162]. In all studies, the goal was to analyze ethanol’s effects on neurodevelopment beyond the period of active exposure to mimic maternal cessation of drinking late in pregnancy or during the early postnatal period. Together, the findings suggest that ethanol-mediated developmental white matter pathology with oligodendrocyte dysfunction progresses over time, well after alcohol exposures have ceased. In other words, the full impact of prenatal alcohol exposure on brain development may not be immediately evident, although the potential for responding to early therapeutic interventions, e.g., choline or dietary soy, likely exists.

Chol and D+G had related but non-identical stimulatory effects on oligodendrocyte myelin protein expression. In general, the responses in paired control and ethanol-exposed cultures were similar, but D+G’s effects on MAG1, MOG, and GALC expression in ethanol-exposed relative to Veh were more robust than Chol’s. This suggests that D+G, in the absence of Chol supplementation, can support MAG1′s role in myelin formation, including myelin-axonal spacing [93,94], and GALC’s role in metabolizing ceramide to sulfatide [100] in brains at risk for FASD-related neurodevelopmental pathologies. The similarly increased levels of MOG and PDGFR-α in Chol- and D+G-treated versus Veh-treated cultures suggest that both types of supplementation support oligodendrocyte maturation and myelination as conferred by MOG [95] and oligodendrocyte progenitor cell survival, growth, and maintenance mediated by PDGFR-α [98]. In contrast, PLP and MBP, the most abundant myelin proteins expressed in mature white matter, were similarly expressed across all cultures and treatments. The statistical differences in PLP (Figure 1D) were due to a modest increase in ethanol–Veh combined with modest reductions in ethanol–Chol, control–D+G, and ethanol–D+G.

Astrocyte and Stress Molecules: The ethanol-associated modest (statistical trend) increase in GFAP expression in Veh-treated cultures corresponds with previous observations in FASD models [45]. This suggests that, in contrast to oligodendrocytes, astrocytic responses occur very early and persist or progress during in vivo development. The significant reductions in GFAP associated with Chol or D+G treatment of ethanol-exposed cultures likely reflect their injury- or harm-reduction effects on ethanol neurotoxicity.

The higher mean level of 8-OHdG in D+G- versus Veh- and Chol-treated cultures, irrespective of ethanol exposure, was of uncertain significance in terms of its effects on oligodendrocyte-myelin proteins and GFAP, since 8-OhdG marks DNA oxidation due to ROS generation [104] and may also signify developmental impairments in growth [103]. On the other hand, the higher level of ubiquitin may paradoxically reflect a positive response to oxidative stress, enabling more efficient removal of unwanted, misfolded proteins that would ultimately exacerbate cellular stress [102]. The significant reductions in HNE detected in ethanol–Chol and ethanol–D+G cultures relative to ethanol–Veh are of further interest given the role of HNE in mediating protein cross-linking, adduct formation, and carbonyl stress [105]. D+G was distinguished by its protective effects in control cultures as well, which correlated with the in vivo finding of enhanced neurodevelopment and function in both control and ethanol-exposed dietary soy-fed animals [70].

Insulin, IGF-1, and IRS1: The analysis of upstream signaling molecules, including levels of total and phosphorylated proteins and the calculated relative phosphorylation (p/T), mainly showed that D+G significantly enhanced signaling through the insulin and IGF-1 receptors relative to Veh but not Chol, and that the effects were similar for control and ethanol-exposed cultures. D+G’s dual enhancement of signaling would support both neuronal and oligodendrocyte functions [109]. Chol’s more modest effects suggest it would be less effective than dietary soy for supporting key signaling pathways utilized in the development of both gray and white matter brain structures.

Akt, GSK-3, and PTEN: Ethanol’s significant inhibitory effect on Akt phosphorylation corresponds with previous in vivo observations [35,41]. The protective effects of Chol were modest but sufficient to normalize responses in the ethanol-exposed animals relative to Veh- and Chol-treated controls. D+G’s more robust effects, manifested by significantly higher levels of ^pS473^-Akt and p/T-Akt in ethanol-exposed cells relative to ethanol–Chol and ethanol–Veh, correspond with the finding that D+G also enhanced signaling through the insulin and IGF-1 receptors in ethanol-exposed cells.

The Chol- and D+G-associated reductions in GSK-3α expression were not accompanied by altered levels of ^pS21^-GSK-3α or p/T-GSK-3α. GSK-3α and GSK-3β have overlapping roles in their regulation of multiple signaling pathways in the CNS [164], including canonical Notch, Wnt/β-catenin, G-coupled receptors, Sonic hedgehog, and receptor tyrosine kinase [165]. The short-term nature of these experiments likely accounts for the discordant findings with respect to earlier reports of impaired Wnt and Notch signaling in the developing brain after chronic in vivo ethanol exposures [57,146,166,167,168]. Concerning GSK-3β, D+G had prominent inhibitory effects on the protein and increased p/T-GSK-3β, which together likely reduced the kinase activity, corresponding with the higher levels of Akt activation. The modest or intermediate rescue effects of Chol again highlight the superior neuroprotection afforded by D+G in this short-term FASD model.

PTEN phosphatase inhibits PI3K-Akt, and CK2 phosphorylation of PTEN slows its proteasome degradation, thereby prolonging its activity [119,120]. The PTEN protein is also stabilized by GSK-3β phosphorylation [118]. The significantly higher levels of PTEN in Chol- and D+G-treated animals relative to Veh appear counterproductive in relation to the protective stimulatory effects the treatments had on Akt (increased p/T-Akt) and inhibitory effects on GSK-3β activity (increased p/T-GSK-3β), particularly in ethanol-exposed cultures. On the other hand, the higher levels of p/T-PTEN in ethanol–Veh relative to control–Veh are consistent with previous reports of increased PTEN activity in the livers and brains of chronic ethanol exposure models [41,169,170]. The significantly reduced levels of p/T-PTEN afforded by Chol or D+G, irrespective of ethanol exposure, may signify higher turnover and a shortened half-life of PTEN, with net reductions in phosphatase activity. However, additional studies are needed to fully interpret the results.

mTOR Signaling: TSC2 is a tumor suppressor gene that encodes tuberin, a growth inhibitory protein and upstream regulator of mTOR [113]. Its interaction with hamartin forms the TSC protein complex that modulates cell growth and negatively regulates mTORC1 signaling [113,124]. Insulin and IGF-1 signaling through Akt inhibit the TSC2:TSC1 complex due to phosphorylation and inactivation of TSC2 [112,123]. The lowest levels of TSC2 protein and highest levels of p/T-TSC2 in D+G-treated cultures correlate with the prominently increased levels of p/T-Insulin-R and p/T-IGF-1 R, reflecting increased receptor kinase activities, and corresponding increases in ^pS473^-Akt, p/T-Akt, and p/T-GSK-3β in ethanol-exposed relative cells to ethanol–Veh. Chol treatment also showed rescue effects concerning p/T-TSC, but to significantly lesser extents than D+G and without significant reductions in either TSC2 protein or ^pS939^-TSC2 expression. In aggregate, these results suggest that while the inhibitory effects of Chol and D+G on TSC2 likely enhanced mTOR signaling, the frontal lobe responses to D+G vis-à-vis ethanol exposure were superior to those of Chol.

mTOR regulates growth and catabolism via multipronged signaling networks linked to two kinase complexes, mTORC1 and mTORC2. mTOR signaling is enabled by its stabilization within mTORC complexes. mTORC1 is formed by mTOR complexing with Raptor, mLST8 (a stabilizer), PRAS40 (an inhibitor), and DEPTOR (an inhibitor). mTORC1 is rapamycin-sensitive and activated by growth factors including insulin and IGF-1, glucose, amino acids, energy, and oxygen, resulting in upregulation of protein synthesis, enhanced lipid synthesis, and mitochondrial biogenesis, along with the inhibition of autophagy through targeted activation of RPS6, stimulation of eIR4b, and inactivation of 4EBP1, an eIF4E inhibitor. Akt phosphorylation of TSC1/2 and PRAS40 releases their inhibitory actions on mTORC1. The resulting activation of p70S6K phosphorylates and activates RPS6 and inhibits eEF2K [171,172,173]. In the brain, mTORC1 signaling promotes cell growth, synaptic plasticity, oligodendrocyte progenitor cell differentiation, and myelination, as well as glial-induced scar formation and gliosis [161,174].

mTORC2 is rapamycin- and nutrient-insensitive but responsive to growth factors that control cell survival, apoptosis, cell proliferation, and cell shape. mTORC2, formed by mTOR complexed with mLST8, Rictor, DEPTOR (inhibitor), mSIN1, and Protor1/2, mediates its effect by phosphorylating and activating Akt, PKC, and SGK1. mTORC2 regulates cell motility, survival, and metabolism, in part through its effect on the actin cytoskeleton. mTORC2 has modest effects on oligodendrocyte differentiation and little effect on myelination [160]. Ser2448 phosphorylation of mTOR by PI3 kinase/Akt signaling enhances binding to Raptor and Rictor proteins and associated mTORC1/2 activities [127,173] and regulates myelination [175].

The reductions in mTOR protein expression measured in both Chol- and D+G-treated control and ethanol-exposed cultures are of uncertain significance. However, the levels of ^pS2448^-mTOR, the active form that binds to Raptor and Rictor in mTORC1/2, were similar among the Veh, Chol, and D+G control cultures and significantly elevated in the ethanol–Chol group relative to all other groups. Importantly, p/T-mTOR was also significantly increased in ethanol–Chol, indicating selectively enhanced signaling through mTORC1 or mTORC2 in ethanol-exposed, Chol-treated frontal lobe cultures. Apart from this exceptional ethanol–Chol result, the mean levels of ^pS2448^-mTOR and p/T-mTOR were similar for all other conditions, suggesting that in general, neither the ethanol exposures nor treatments with Chol or D+G significantly impacted mTOR’s capacity to interact with Raptor or Rictor in mTORC1/2 complexes.

p70S6K is a major downstream target of mTORC1 signaling and, when activated, phosphorylates RPS6 [128]. The progressive declines in p70S6K and RPS6 proteins from Veh to Chol to D+G parallel the trends observed for mTOR. However, the somewhat reciprocal trend-wise increases in p/T-p70S6K likely reflect higher levels of p70S6K activity corresponding to enhanced mTORC1 signaling with Chol or D+G treatment relative to Veh in both control and ethanol-exposed cultures.

RPS6 phosphorylation and kinase activation by p70S6K reflect downstream mTORC1 signaling [130]. The significantly higher levels of ^pS235/236^-RPS6 and p/T-RPS6, i.e., kinase activation, in both control–Chol and ethanol–Chol samples versus Veh or D+G treatment suggest that Chol exerts superior actions on mTORC1-mediated cell survival and growth while inactivating BAD, a pro-apoptotic molecule [128]. The long-term impact of Chol’s stimulatory effects on downstream mTORC1 signaling requires further investigation with in vivo models of FASD.

## 4. Summary

In this short-term model, the main significant ethanol effects were associated with alterations in MAG1, HNE, p-IGF1-R, p-Akt, PTEN, p/T-PTEN, p/T-TSC2, p-mTOR, and p/T-mTOR, and statistical trend effects occurred with respect to p-Insulin R, p/t-IGF-1 R, p/T-Akt, p/t-GSK-3β, and p/T-P70S6K.

Both Choline and D+G soy isoflavone treatments broadly impacted myelin protein expression, although their effects were non-identical and generally more robust for D+G than Choline.Choline and D+G treatments significantly affected both control and ethanol-exposed cultures.Ethanol-associated astrocyte activation (increased GFAP) was similarly suppressed by Choline and D+G soy isoflavones, but lipid peroxidation was more effectively dampened by D+G than Choline.D+G soy isoflavones enhanced signaling through both the insulin and IGF-1 receptors, whereas Choline significantly impacted IGF-1 and not insulin receptor tyrosine kinase signaling.Both the breadth and magnitude of enhanced Akt/mTOR signaling were greater with D+G than Choline relative to Vehicle.A unique effect of Choline was to significantly upregulate pS235/236-RPS6 and p/T-RPS6 relative to Veh and D+G, suggesting superior end results with respect to mTOR pathway activation via p70S6K.

## 5. Materials and Methods

### 5.1. Reagents and Resources

Choline chloride was purchased from Thermo Fisher (Waltham, MA, USA). Daidzein (>98% purity) and Genistein (>95% purity) were purchased from Sigma-Aldrich Company (St. Louis, MO, USA). Commercial antibodies, including their sources, concentrations or dilutions used, vendors, and RRID numbers, are listed in Table 5. Bicinchoninic acid (BCA) reagents, horseradish peroxidase (HRP)-conjugated secondary antibody, superblock (TBS), and enzyme-linked immunosorbent assay (ELISA). MaxiSorp 96-well plates were purchased from Thermo Fisher Scientific (Bedford, MA, USA). Amplex red soluble fluorophore and 4-Methylumbelliferyl phosphate (4-MUP) were purchased from Life Technologies (Carlsbad, CA, USA). Alkaline Phosphatase Streptavidin and the Proton Biotin Protein Labeling Kit were purchased from Vector Laboratories Inc. (Newark, CA, USA). Total and Phospho-Akt/mTOR Pathway panels were purchased from MilliporeSigma (Bedford, MA, USA). The Luminex MAGPIX system was purchased from Luminex Corp. (Austin, TX, USA). The SpectraMax M5 microplate reader was purchased from Molecular Devices Corp. (Sunnyvale, CA, USA).

### 5.2. Experimental Model

The use of rats for this research was approved by the Lifespan Institutional Animal Care and Use Committee (IACUC), Board Reference #500221. Six Long Evans rat litters were used for these experiments. Three litters each received intraperitoneal (i.p.) injections of saline (control) or ethanol (2 g/kg in saline) in 50 µL on postnatal days P3 and P5. These binge treatments produced blood alcohol levels between 193.1 and 331.6 mg/dL versus 2.4 to 11.9 mg/dL in the saline-injected controls. On P7, the rats were sacrificed to generate frontal lobe slice cultures (FLSCs) as described [61,140,141]. In brief, freshly harvested frontal lobes were chilled in ice-cold Hank’s balanced salt solution (HBSS) and sliced at 250 μm intervals using a McIlwain tissue chopper (Mickle Laboratory Engineering Co. Ltd., Guildford, Surrey, UK). The slices were separated under a dissecting microscope and placed into 12-well Nunc plates (4 or 5 slices per well). The slices from each litter were evenly divided for the three within-group (control versus ethanol) treatment conditions (Vehicle, Choline chloride, or Daidzein+Genistein). The cultures were maintained in Dulbecco’s Modified Eagles Medium (DMEM) supplemented with 10% heat-inactivated fetal bovine serum (FBS), 4 mM L-glutamine, 8.5 g/L glucose, 25 mM potassium chloride, 120 U/mL penicillin, 120 µg/mL streptomycin, and 1X MEM non-essential amino acid solution at 37 °C in standard 5% CO_2_ incubators without further ethanol exposures. Both male and female frontal lobe slices were included in all culture conditions. Sex was not investigated as a biological variable because in previous studies, sex differences in molecular and biochemical responses to alcohol or dietary soy were not observed, despite differences in growth, body weight, and organ weights [70,71].

The cultures were treated with vehicle (Veh), 75 µM Choline Chloride (Chol), or 1 µM Daidzein + 1 µM Genistein (D+G) for 72 h with daily medium changes for fresh addition of supplements. The Chol [180,181,182,183] and D+G [184,185] doses were based on previous reports together with dose optimization to ensure absence of cytotoxicity based on viability assays. Since the Chol, Daidzein, and Genistein stock solutions were prepared in dimethylsulfoxide (DMSO), the Veh treatments included 0.02% DMSO in serum-free medium to match the final concentration of DMSO in the diluted Chol or D+G solutions. At the conclusion of the experiment, the cultured tissue slices were harvested and homogenized in a proprietary phosphatase inhibitor-containing buffer (MilliporeSigma, Bedford, MA, USA) supplemented with a protease inhibitor cocktail to include 1 mM PMSF, 0.1 mM TPCK, 2 µg/mL aprotinin, 2 µg/mL pepstatin A, and 1 µg/mL leupeptin. The supernatant fractions obtained after centrifuging the homogenates at 14,000 rpm for 15 min at 4 °C were used in duplex ELISAs and Akt/mTOR 11-Plex Multiplex assays. Protein concentrations were measured with the bicinchoninic acid (BCA) assay.

### 5.3. Duplex ELISAs

Duplex ELISAs measured immunoreactivity to myelin-associated glycoprotein 1 (MAG), myelin oligodendrocyte glycoprotein (MOG), myelin basic protein (MBP), proteolipid protein (PLP), galactosylceramidase (GALC), platelet-derived growth factor receptor-alpha (PDGFR-α), glial fibrillary acidic protein (GFAP), ubiquitin (UBQ), 8-Hydroxyguanosine (8-OHDG), and 4-hydroxy-2-nonenal (HNE) with results were normalized to large acidic ribosomal protein (RPLPO) as previously described [57,61,70]. In brief, duplicate 50 μL aliquots containing 50 ng of protein were adsorbed to the bottoms of 96-well MaxiSorp plates by overnight incubation at 4 °C. Non-specific sites were blocked with the Superblock (TBS) blocking buffer. Proteins were reacted with primary antibodies (0.2–5.0 μg/mL) overnight at 4 °C. Immunoreactivity was detected with HRP-conjugated secondary antibodies and the Amplex UltraRed soluble fluorophore. Fluorescence was measured in a SpectraMax M5 microplate reader (*Ex530 nm/Em590 nm*). Then, RPLPO immunoreactivity, which served as a sample loading control, was measured in the same plates by incubating the proteins with biotinylated anti-RPLPO followed by streptavidin-conjugated alkaline phosphatase and 4-MUP (*Ex360 nm/Em450 nm*). The calculated ratios of the target protein to RPLPO fluorescence were used for inter-group comparisons. Six replicate cultures were analyzed per group.

### 5.4. Multiplex ELISAs

Commercial bead-based Total and Phospho-Akt/mTOR Magnetic 11-Plex panels were used to examine the effects of ethanol and treatment with Chol or D+G on the expression and phosphorylation of proteins integrally related to insulin and IGF-1 signaling through the Akt and mTOR pathways (MilliporeSigma, Bedford, MA, USA). The Total Akt/mTOR panel measured immunoreactivity to the insulin receptor (Insulin R), IGF-1 receptor (IGF-1R), insulin receptor substrate, type 1 (IRS-1), Akt, phosphatase and tensin homolog (PTEN), glycogen synthase kinase 3α (GSK-3α), glycogen synthase kinase 3β (GSK-3β), tuberous sclerosis protein 2 (TSC2), mammalian target of rapamycin (mTOR), ribosomal protein S6 kinase beta-1 (p70S6K), and ribosomal protein S6 (RPS6). The Phospho-Akt/mTOR panel measured immunoreactivity to ^pYpY1162/1163^-Insulin R, ^pYpY1135/1136^-IGF-1R, ^pS636^-IRS-1, ^pS473^-Akt, ^pS380^-PTEN, ^pS21^-GSK3α, ^pS9^-GSK3β, ^pS939^-TSC2, ^pS2448^-mTOR, ^pT412^-p70S6K, and ^pS235/236^-RPS6 according to the manufacturer’s protocol. FLSC tissue sample homogenates containing 12.5 μg protein were incubated with antibody-bound beads. Captured antigens were detected with biotinylated secondary antibodies and phycoerythrin-conjugated streptavidin. Fluorescence intensity was measured with a MAGPIX. Data are expressed as fluorescence light units (FLU).

### 5.5. Statistics

Statistical analyses were used to examine the very early effects of rat postnatal (3rd trimester human equivalent) binge alcohol exposures on glial and stress protein expression and signaling through Akt-mTOR pathways in a model that was shown to produce sustained FASD effects [37,61,87,88,89,90]. In addition, the analyses were used to assess the very early effects of treatment interventions that are known to both enhance normal brain development and remediate neurobehavioral, neuropathological, and neurobiochemical abnormalities, including FASD, to better understand their mechanisms of action [64,70,71,73,74,75,80,86]. Inter-group comparisons were made using two-way analysis of variance (ANOVA) with Tukey post-hoc multiple comparison tests and a 5% false discovery rate (GraphPad Prism 9.4, San Diego, CA, USA). The F-ratios, degrees of freedom, and *p*-values with significant (*p* < 0.05) or trend-wise (0.05 < *p* < 0.10) differences highlighted are shown in the Tables. The significant post-hoc test results are depicted in the graphs.

## 6. Conclusions

D+G soy isoflavones, without supplemental Choline, exerted superior support for oligodendrocyte myelin protein expression and inhibition of pro-inflammatory/oxidative stress mediators in immature frontal lobe tissue compared with Veh and Chol.Relative to Veh, the comparable levels of support afforded by Choline or D+G soy isoflavones in control versus ethanol-exposed cultures highlight their potential benefits for enhancing neurodevelopment independent of prenatal ethanol exposure, i.e., FASD.The aggregate results support the concept that FASD preventive or harm-reduction approaches should include encouraging the consumption of dietary soy, which is naturally rich in choline and, together with its isoflavones, would likely be more effective than Choline micronutrient supplementation.Alternatively, future approaches for FASD may include treatment with novel pharmaceutical isoflavones that are neuroprotective and support signaling pathways that promote neuroplasticity, including mTOR [149].A limitation of this study is its short-term design with data capture prior to the full development of FASD-associated cellular and molecular pathologies pertaining to impairments in myelin protein expression and insulin/IGF-1 signaling through Akt pathways. Follow-up experiments to compare long-term in vivo responses to soy protein isolate and choline in an FASD model will address this limitation.

## Figures and Tables

**Figure 1 ijms-24-07595-f001:**
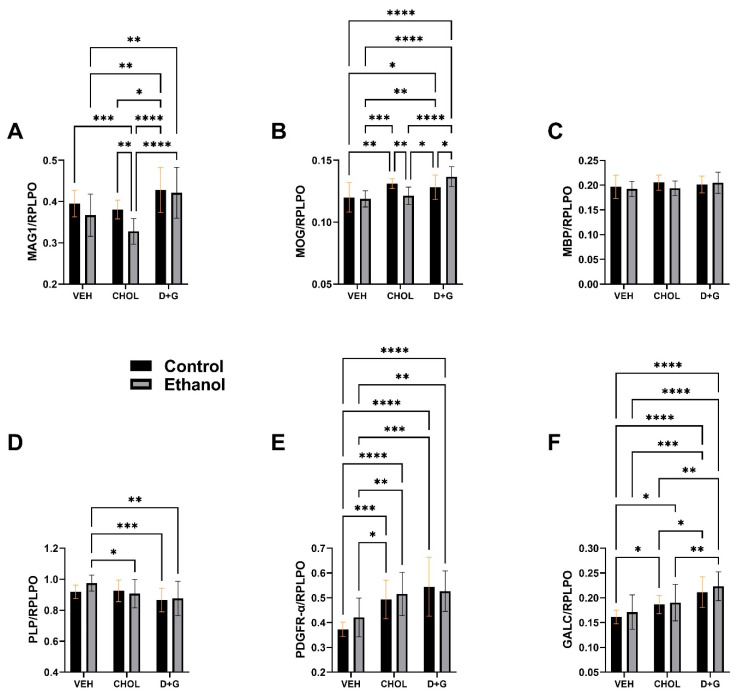
Ethanol and treatment (Chol and D+G) effects on (**A**) MAG1, (**B**) MOG, (**C**) MBP, (**D**) PLP, (**E**) PDGFR-α, and (**F**) GALC in rat frontal lobe slice cultures. Immunoreactivity was measured by duplex ELISA, with results normalized to RPLPO. Graphs depict the mean ± SD of results (N = 6 cultures/group). Inter-group differences were analyzed by two-way ANOVA (see Table 1) with post-hoc multiple comparisons using Tukey tests. * *p* < 0.05; ** *p* < 0.01; *** *p* < 0.001; **** *p* < 0.0001.

**Figure 2 ijms-24-07595-f002:**
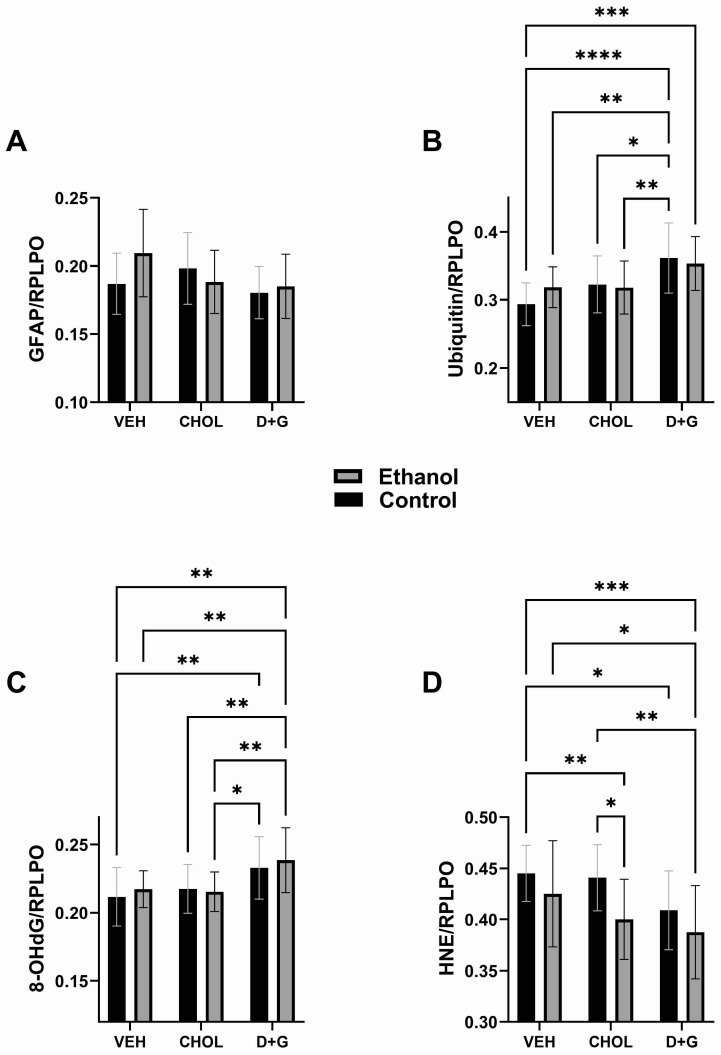
Ethanol and treatment (Chol and D+G) effects on (**A**) GFAP, (**B**) Ubiquitin, (**C**) 8-OHdG, and (**D**) HNE measured by duplex ELISA. Graphs depict the mean ± SD of immunoreactivity (N = 6 cultures/group). Inter-group differences were analyzed by two-way ANOVA (see Table 1) with post-hoc multiple comparisons using Tukey tests. * *p* < 0.05; ** *p* < 0.01; *** *p* < 0.001; **** *p* < 0.0001.

**Figure 3 ijms-24-07595-f003:**
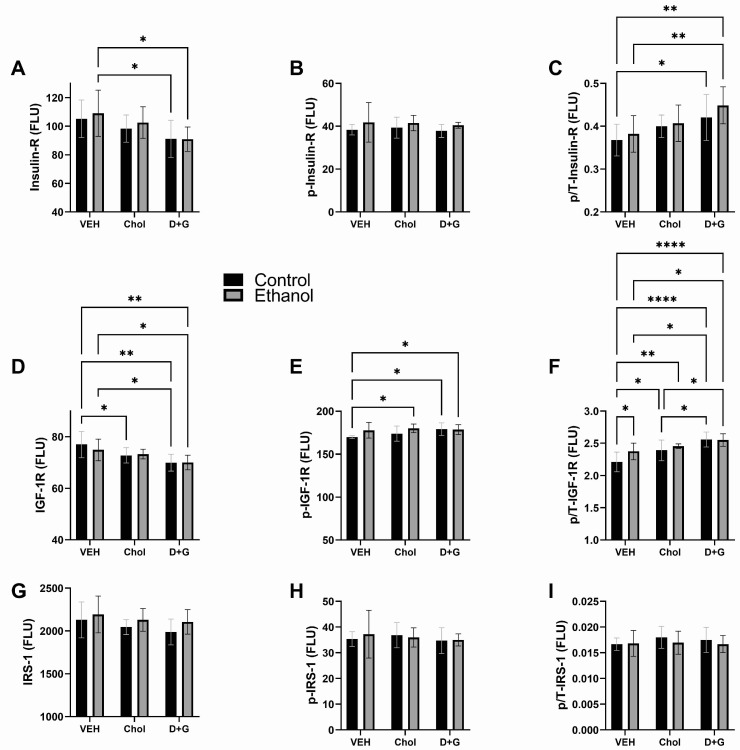
Ethanol and treatment (Chol and D+G) effects on upstream Insulin/IGF-1-Akt-mTOR pathway molecules. (**A**) Insulin R, (**B**) ^pYpY1162/1163^-Insulin-R, (**C**) p/T-Insulin-R, (**D**) IGF-1R, (**E**) ^pYpY1135/1136^-IGF-1R, (**F**) p/T-IGF-1R, (**G**) IRS1, (**H**) ^pS636^-IRS-1, and (**I**) p/T-IRS1. Graphs depict the mean ± SD of results (N = 6 cultures/group; FLU = fluorescent light units). Inter-group differences were analyzed by two-way ANOVA (see Table 2) with post-hoc multiple comparisons using Tukey tests. * *p* < 0.05; ** *p* < 0.01; **** *p* < 0.0001.

**Figure 5 ijms-24-07595-f005:**
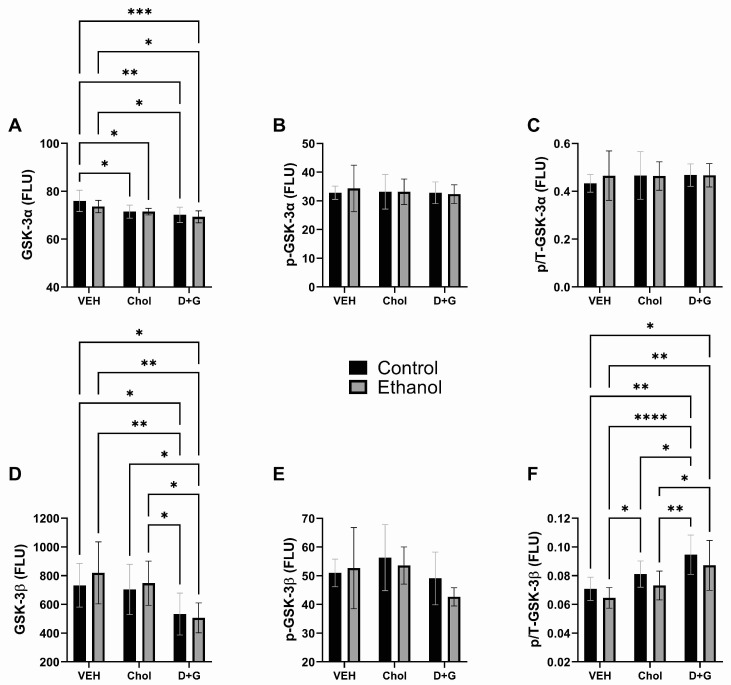
Ethanol and treatment **(**Chol and D+G) effects on intermediate key signaling molecules in the Insulin/IGF-1-Akt-mTOR pathway. (**A**) GSK-3α, (**B**) ^pS21^-GSK-3α, (**C**) p/T-GSK-3α, (**D**) GSK-3β, (**E**) ^pS9^-GSK-3β, and (**F**) p/T-GSK-3β. Graphs depict the mean ± SD of immunoreactivity (N = 6 cultures/group; FLU = fluorescent light units). Inter-group differences were analyzed by two-way ANOVA (see Table 3) with post-hoc Tukey tests. * *p* < 0.05; ** *p* < 0.01; *** *p* < 0.001; **** *p* < 0.0001.

**Figure 6 ijms-24-07595-f006:**
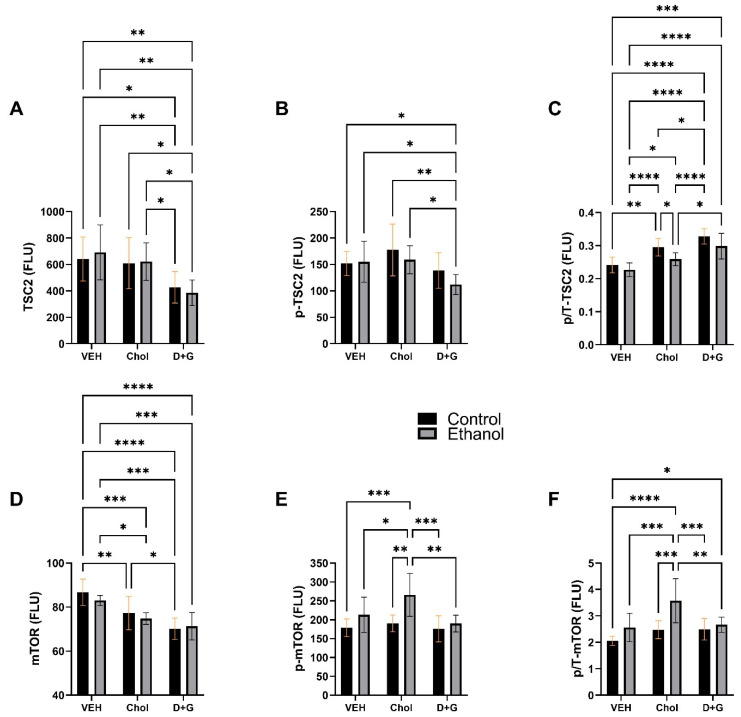
Ethanol and treatment **(**Chol and D+G) effects on the downstream signaling molecules in the Insulin/IGF-1-Akt-mTOR pathway. (**A**) TSC2, (**B**) ^pS939^-TSC2, (**C**) p/T-TSC2, (**D**) mTOR, (**E**) ^pS2448^-mTOR, (**F**) p/T-mTOR. Graphs depict the mean ± SD of immunoreactivity (N = 6 cultures/group; FLU = fluorescent light units). Inter-group differences were analyzed by a two-way ANOVA (see Table 4) with post-hoc multiple comparisons using Tukey tests. * *p* < 0.05; ** *p* < 0.01; *** *p* < 0.001; **** *p* < 0.0001.

**Figure 7 ijms-24-07595-f007:**
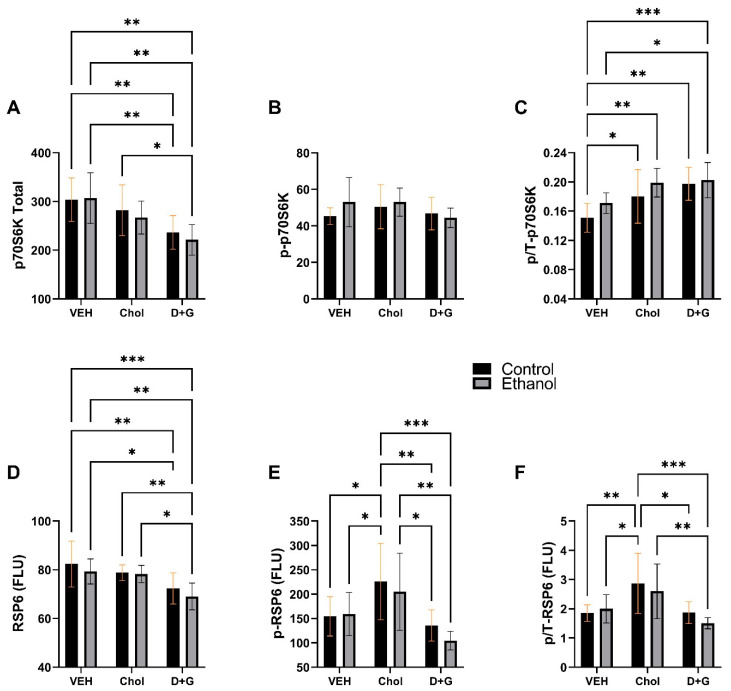
Ethanol and treatment **(**Chol and D+G) effects on key downstream signaling molecules in the Insulin/IGF-1-Akt-mTOR pathway. (**A**) p70S6K, (**B**) ^pT412^-p70S6K, (**C**) p/T-p70S6K, (**D**) RPS6, (**E**) ^pS235/236^-RPS6, (**F**) p/T-RPS6. Graphs depict the mean ± SD of results (N = 6 cultures/group; FLU = fluorescent light units). Inter-group differences were analyzed by a two-way ANOVA (see Table 4) with post-hoc multiple comparisons using Tukey tests. * *p* < 0.05; ** *p* < 0.01; *** *p* < 0.001.

**Table 1 ijms-24-07595-t001:** Effects of In Vivo Ethanol Exposure and Ex Vivo Interventive Treatments on Glial and Stress Molecule Expression in Frontal Lobe White Matter Slice Cultures.

Molecule	Ethanol Factor		Treatment Factor		Ethanol x Treatment Interaction	
	F-Ratio	*p*-Value	F-Ratio	*p*-Value	F-Ratio	*p*-Value
MAG1	**7.86**	**0.0066**	**15.30**	**<0.0001**	1.589	N.S.
MOG	0.161	N.S.	**16.11**	**<0.0001**	**7.52**	**0.001**
MBP	0.925	N.S.	1.45	N.S.	1.06	N.S.
PLP	0.817	N.S.	**5.88**	**0.0045**	1.46	N.S.
PDGFR-α	0.811	N.S.	**18.33**	**<0.0001**	0.968	N.S.
GALC	1.574	N.S.	**19.67**	**<0.0001**	0.125	N.S.
GFAP	0.979	N.S.	** *2.41* **	** *0.0974* **	* **2.55** *	** *0.0855* **
UBIQUITIN	0.192	N.S.	**10.81**	**<0.0001**	1.269	N.S.
8-OHdG	0.446	N.S.	**8.798**	**<0.0001**	0.320	N.S.
HNE	**8.464**	**0.0049**	**5.167**	**0.0082**	0.506	N.S.

Duplex ELISAs measured immunoreactivity in control and ethanol-exposed rat frontal lobe slice culture samples (50 ng/sample; 6 rats per group). The table lists the two-way ANOVA test results (F-Ratios and *p*-Values) for ethanol exposure, treatment (Chol or D+G), and ethanol x treatment interactive effects on glial and stress molecule expression. The bold (non-italicized) font highlights significant results. Bold italics highlight statistical trends. N.S. = not statistically significant; DF for all tests: interaction DFn, DFd (2, 30); treatment DFn, DFd (2, 30); exposure DFn, DFd (1, 30).

**Table 2 ijms-24-07595-t002:** Effects of In Vivo Ethanol Exposure and Ex Vivo Interventive Treatments on Insulin, IGF-1, and IRS-1 Signaling Mediators in Frontal Lobe Slice Cultures.

Index	Ethanol Factor		Treatment Factor		Ethanol x Treatment Interaction	
	F-Ratio	*p*-Value	F-Ratio	*p*-Value	F-Ratio	*p*-Value
Insulin R	0.423	N.S.	**5.272**	**0.011**	0.127	N.S.
p-Insulin R	** *3.014* **	** *0.093* **	0.182	N.S.	0.059	N.S.
p/T-Insulin R	1.417	N.S.	**6.144**	**0.006**	0.192	N.S.
IGF-1 R	0.182	N.S.	**8.504**	**0.001**	0.467	N.S.
p-IGF-1 R	**4.204**	**0.049**	1.731	N.S.	1.257	N.S.
p/T-IGF-1 R	** *3.285* **	** *0.080* **	**13.85**	**<0.0001**	1.507	N.S.
IRS-1	2.694	N.S.	1.512	N.S.	0.081	N.S.
p-IRS-1	0.066	N.S.	0.325	N.S.	0.213	N.S.
p/T-IRS-1	0.638	N.S.	0.388	N.S.	0.279	N.S.

Immunoreactivity was measured with 11-Plex Akt/mTOR total protein and phosphoprotein magnetic bead-based assays. The Table lists the two-way ANOVA test results (F-Ratios and *p*-Values) for ethanol exposure, treatment (Chol or D+G), and ethanol x treatment interactive effects on Insulin R, IGF-1R, IRS-1, ^pYpY1162/1163^-Insulin R, ^pYpY1135/1136^-IGF-1R, and ^pS636^-IRS-1, and the calculated relative phosphorylation (p/T) levels of each protein are also listed. DF for all tests: interaction DFn, DFd (2, 30); treatment DFn, DFd (2, 30); exposure DFn, DFd (1, 30). The bold (non-italicized) font highlights significant results. Bold italics highlight statistical trends. Abbreviations: R = receptor; p = phosphorylated; N.S. = not statistically significant. N = 6 rats/group with 3 replicate cultures per rat.

**Table 4 ijms-24-07595-t004:** Effects of In Vivo Ethanol Exposure and Ex Vivo Interventive Treatments on TSC2, mTOR, p70S6K, and RPS6 Signaling Mediators in Frontal Lobe Slice Cultures.

Protein	Ethanol Factor		Treatment Factor		Ethanol x Treatment Interaction	
	F-Ratio	*p*-Value	F-Ratio	*p*-Value	F-Ratio	*p*-Value
TSC2	0.01	N.S.	**9.036**	**0.0008**	0.249	N.S.
p-TSC2	1.568	N.S.	**5.174**	**0.012**	0.657	N.S.
p/T-TSC2	**9.47**	**0.004**	**27.27**	**<0.0001**	0.549	N.S.
mTOR	0.899	N.S.	**21.86**	**<0.0001**	0.682	N.S.
p-mTOR	**11.27**	**0.002**	**4.800**	**0.016**	2.187	N.S.
p/T-mTOR	**13.59**	**0.0009**	**6.801**	**0.0037**	* **2.854** *	* **0.073** *
P70S6K	0.137	N.S.	**9.740**	**0.0006**	0.192	N.S.
p-P70S6K	0.72	N.S.	1.337	N.S.	0.872	N.S.
p/T-P70S6K	** *3.361* **	** *0.077* **	**8.609**	**0.0011**	0.368	N.S.
RPS6	0.407	N.S.	**9.917**	**0.0005**	0.189	N.S.
p-RPS6	0.763	N.S.	**9.538**	**0.0002**	0.354	N.S.
p/T-RPS6	0.587	N.S.	**8.952**	**0.0009**	0.549	N.S.

Immunoreactivity was measured with 11-Plex Akt/mTOR total protein and phosphoprotein magnetic bead-based assays. The table lists the two-way ANOVA test results for in vivo ethanol, ex vivo treatment (Chol or D+G), and ethanol x treatment interactive effects on TSC2, mTOR, p70S6K, RPS6, ^pS939^-TSC2, ^pS2448^-mTOR, ^pT412^-p70S6K, and ^pS235/236^-RPS6, and the calculated relative phosphorylation levels (p/T). DF for all tests: interaction DFn, DFd (2, 30); treatment DFn, DFd (2, 30); exposure DFn, DFd (1, 30). The bold font highlights significant results. Bold italics highlight statistical trends. Abbreviations: R = receptor; p = phosphorylated; N.S. = not statistically significant. N = 6 rats/group with 3 replicate cultures per rat.

**Table 5 ijms-24-07595-t005:** Antibodies Used for Duplex ELISA Studies.

Antibody	Source	Type	Stock	Final Concentration	Commercial Source	RRID#
Myelin-associated glycoprotein (MAG)	Mouse	Monoclonal	0.5 mg/mL	0.25 µg/mL	Abcam Biotechnology, Waltham, MA, USA	AB_2042411
Myelin Oligodendrocyte Glycoprotein (MOG)	Rabbit	Polyclonal	1.0 mg/mL	1.25 µg/mL	Abcam Biotechnology, Waltham, MA, USA	AB_2145529
Myelin Basic Protein (MBP)	Rabbit	Polyclonal	1.0 mg/mL	5 µg/mL	Sigma-Aldrich, St. Louis, MO, USA	AB_1841021
Proteolipid Protein (PLP)	Rabbit	Polyclonal	Purified serum	1:2000	Abcam Biotechnology, Waltham, MA, USA	AB_776593
Galactosylceramidase (GALC)	Rabbit	Polyclonal	1.0 mg/ml	2 µg/mL	Abcam Biotechnology, Waltham, MA, USA	AB_2108528
Platelet-Derived Growth Factor Receptor-Alpha (PDGFR-α)	Rabbit	Polyclonal	1.0 mg/mL	1 µg/mL	Abcam Biotechnology, Waltham, MA, USA	AB_2162341
Glial Fibrillary Acidic Protein (GFAP)	Mouse	Monoclonal	1.0 mg/mL	2.5 µg/mL	Invitrogen (Life Technologies), Waltham, MA, USA	AB_2535827
Ubiquitin (UBQ)	Rabbit	Polyclonal	0.25 mg/mL	0.5 µg/mL	Abcam Biotechnology, Waltham, MA, USA	AB_306069
8-Hydroxyguanosine (8-OHdG)	Mouse	Monoclonal	1.0 mg/mL	0.2 µg/mL	Abcam Biotechnology, Waltham, MA, USA	AB_867461
4-Hydroxynonenal (4-HNE)	Goat	Polyclonal	0.8 mg/mL	1.0 µg/mL	Abcam Biotechnology, Waltham, MA, USA	AB_722493
Large acidic ribosomal protein (RPLPO)	Mouse	Monoclonal	0.1 mg/mL	0.1 µg/mL	Santa Cruz, Dallas, TX, USA	[176,177,178,179]

## Data Availability

Not applicable.

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
