# Peer review of "Differential Early Mechanistic Frontal Lobe Responses to Choline Chloride and Soy Isoflavones in an Experimental Model of Fetal Alcohol Spectrum Disorder"

_ijms, 2023, doi:10.3390/ijms24087595_

Round 1

Reviewer 1 Report

This study by Tong et al investigated the effects of choline (Chol) and the Daidzein+Genistein (D+G) soy isoflavones on rat frontal lobe tissue slices exposed to alcohol in vivo. In their experimental model, the rat pups were binge administered 2g/Kg of ethanol or saline (control) on postnatal days (P) 3 and P5. After that P7 frontal lobe slice cultures were treated with vehicle (Veh), Choline chloride (Chol; 75µM), or D+G (1µM each) for 72 h further ethanol exposures. They examined the expression levels of myelin oligodendrocyte proteins and stress-related molecules and Akt-mTOR signaling in the slice cultures. They found D+G and Chol significantly enhanced the expression of several oligodendrocyte myelin proteins and reduced GFAP and 4-hydroxynonenal relative to controls, but the responses were more robust with D+G. Although most mediators of Akt-mTOR signaling were better supported by D+G than Chol, RPS6 phosphorylation was significantly increased by Chol and not D+G relative to controls. Both Chol and D+G stimulated the expression of oligodendrocyte myelin proteins and supported signaling through Akt-mTOR/mTORC1 pathways, but the effectiveness was generally greater with D+G. The study has some potential therapeutic implication. However, the data are mostly descriptive. This reviewer has a major concern regarding the relevance of this experimental model. Specific comments are provided below:

Major concerns:

1.    The authors need to provide a justification of the relevance of this model to FASD. Generally, there was no difference between control and alcohol-treated groups. This raises a concern of usefulness of this model. For example, for IGF-1 signaling, the ethanol factor is mostly insignificant (Table 3A).

2.    Switching between in vivo and in vitro experiments created many confounding conditions. It is better to stick to one condition, such as performing all studies in rat pups in vivo.

3.    In the introduction, the authors need to describe the constituents of soy, such as, what is the percentage of choline and soy isoflavones?

4.    Please provide a justification how the concentrations of 75 μM Chol, and 1 μM Daidzein + 1 μM Genistein (D+G) were selected and their physiological relevance.

5.    The analyses were diffusive and confusing. To this reviewer, the analyses need to be focused on the signaling components that alcohol had impact on, such as p/T-IGF-1R, p-AKT, p/T-PTEN.

Minor concerns:

1. In graph 7D, 7E and 7F, the label of y axis should be RPS6.

2. Please correct the spelling of RPS6 in line 474, 513, 515 and 757.

Reviewer 2 Report

This manuscript describes the effects of acute choline or isoflavones on a number of proteins in ex vivo brain preparations obtained from rats exposed to developmental alcohol, to determine how these two potential interventions may impact brain development. This study is novel and of interest to readers. However, there are several issues that should be addressed.

1.       What was the rationale for the alcohol exposure model, with exposure on 2 postnatal days? Also, i.p. injection can produce additional stress responses, so why that exposure paradigm?

2.       Relatedly, please provide some rationale for examining the acute actions of these nutritional interventions. As noted in the discussion, it is possible that the effects on are not long-lasting or translate to long-term changes in white matter and it is not clear how the acute actions translate to findings where an intervention is administered chronically. Why not treat the subjects with choline and/or D+G in vivo for a longer period of time? Please add more to the discussion about this limitation.

3.       Another limitation is that only one dose of each intervention was applied. Given that the goal was to compare the two interventions, it is possible that any differential effects are related to dose. How were the doses chosen to be somewhat comparable? Again, this limitation needs to be discussed.

4.       What were the blood alcohol concentrations?

5.       How many litters were used for subject generation? It is important to control for potential litter effects, but it is not clear from the methods section is this was taken into account.

6.       Was sex included as a factor in the analyses?  Please add information regarding sex—were only males examined? Given potential sex differences, this is an important factor to consider and should be included in the analyses.

7.       One major issue relates to the statistical analyses. It appears that all potential post hoc comparisons were conducted regardless of statistical outcome. Many of these comparisons are not justified and the authors should conduct only appropriate post hoc analyses to follow-up significant main or interactive effects (in other words, if there is no significant interaction of treatment x ethanol, there is not a justification to make all possible group comparisons). Making every possible comparison greatly increases the likelihood of Type 1 errors. Relatedly, it is fine to examine trends, but 0.09 follow-ups are stretching it (GFAP). The statistics and results need to be modified accordingly.

8.       Relatedly, some graphs show significant differences when there are no significant effects in the overall analyses (Fig. 3G). Moreover, the symbols in the figures should be simplified, illustrating only the justified comparisons.

Minor corrections/clarifications:

-Estimates for FASD in the US may be higher than what is reported (see Phil May’s estimates from school age children).

-What neurobehavioral effects are improved with soy supplementation?

-Line 97: Not sure those references refer to the corpus callosum

-The figure captions don’t need to repeat all of the methods. This is redundant and should be edited.

Round 2

Reviewer 1 Report

The authors have addressed my comments.